# Investigation into Mode Localization of Electrostatically Coupled Shallow Microbeams for Potential Sensing Applications

**DOI:** 10.3390/mi13070989

**Published:** 2022-06-24

**Authors:** Ayman M. Alneamy, Hassen M. Ouakad

**Affiliations:** 1Department of Mechanical Engineering, Jazan University, Jazan 45142, Saudi Arabia; alneamy@jazanu.edu.sa; 2Department of Mechanical and Industrial Engineering, Sultan Qaboos University, Al-Khoudh, P.O. Box 33, Muscat 123, Oman

**Keywords:** MEMS, veering, mode-localization, snap-through, crossing

## Abstract

With the constant need for the development of smart devices, Micro-Electro-Mechanical Systems (MEMS) based smart sensors have been developed to detect hazard materials, micro-particles or even toxic substances. Identifying small particles using such micro-engineering technology requires designing sensors with high sensitivity, selectivity and ease of integration with other electronic components. Nevertheless, the available detection mechanism designs are still juvenile and need more innovative ideas to be even more competitive. Therefore, this work aims to introduce a novel, smart and innovative micro-sensor design consisting of two weakly electrostatically coupled microbeams (both serving as sensors) and electrically excited using a stationary electrode assuming a dc/ac electric signal. The sensor design can be tuned from straight to eventually initially curved microbeams. Such an arrangement would develop certain nonlinear phenomena, such as the snap-through motion. This behavior would portray certain mode veering/mode crossing and ultimately mode localization and it would certainly lead in increasing the sensitivity of the mode-localized based sensing mechanism. These can be achieved by tracking the change in the resonance frequencies of the two microbeams as the coupling control parameter is varied. To this extent, a nonlinear model of the design is presented, and then a reduced-order model considering all geometric and electrical nonlinearities is established. A Long-Time Integration (LTI) method is utilized to solve the static and dynamics of the coupled resonators under primary lower-order and higher-order resonances, respectively. It is shown that the system can display veering and mode coupling in the vicinity of the primary resonances of both beams. Such detected modal interactions lead to an increase in the sensitivity of the sensor design. In addition, the use of two different beam’s configurations in one device uncovered a possibility of using this design in detecting two potential substances at the same time using the two interacting resonant peaks.

## 1. Introduction

Micro-Electro-Mechanical Systems (MEMS) designate a smart and innovative technology integrating small devices with a combination of miniature mechanical and electrical components. Most of these devices are fabricated using bulk micro-machining, surface micro-machining and lithography processes and their assembly is based on integrating semiconductors’ circuitry processes, such as PolyMUMPs, SOIMUMPs and PiezoMUMPs [1,2,3,4]. These tiny devices are used principally as either sensors or actuators and are central to the revolution of the latest industrial technologies due to their main advantages, such as their small size and their low cost of production. Furthermore, these devices can provide significant improvements in terms of performances compared to large macro-scale devices. Due to the diversity of their application fields, they are prominent in a variety of industries, such as medical applications, automobile industries and RF communications.

There are different methods to excite MEMS devices, such as electrothermal, electromagnetic, piezoelectric and electrostatic. Among these methods, the electrostatic actuation has been developed and utilized in a wide variety of applications due to its many inherent advantages. For example, it offers good coupling between different energy domains in the micro-scale, requires small current compared to other actuation methods, reduces the overall power consumption, produces a controllable distance and requires a simple fabrication process [2,3,4]. The electrostatic output force is relatively small compared to other actuation methods but it results in a simple design that involves only two electrodes, one of which is fixed, called the stationary electrode, and another that is able to move toward or away from the fixed electrode depending on the adopted actuation mechanism [5].

The electrostatic actuation has several arrangements, such as parallel electrodes (microbeams) based capacitor and even interdigitated comb-fingers, where in both, the force can be attractive when the two electrodes are of opposite charges and repulsive when they are carrying the same electrical charges. However, this actuation mechanism results in some nonlinear phenomena, such as dynamic pull-in instability where the electrostatic actuating force becomes higher than restoring mechanical force and limits the travel ranges of MEMS devices [6,7,8,9]. This pull-in instability is a major challenge in the design of electrostatically actuated microbeams where they exhibit only one stable equilibrium and beyond which the micro-structure loses its structural stability [10].

To overcome this challenge and to increase the electrostatically actuated micro-structures travel range, flexible and slightly curved beams (shallow arches) have been suggested and then thoroughly examined [11,12,13,14,15]. Indeed, the slightly curved microbeams have the ability to move from one stable equilibrium to another stable equilibrium or oscillate under the same excitation load. The transition between two stable points is commonly referred to as snap-through motion whereas the structure is called bistable [11,12]. Curved microbeams can be fabricated by buckling straight beams through compressive axial loads (buckled beams) or intentionally designed to be curved [16]. MEMS bistable structures have been used in a variety of applications including micro-valves, electrical micro-relays, mechanical memories, gas sensors switches and tweezers [5,17].

Due to the interaction between the geometry nonlinearity (mid-plane stretching) and the force nonlinearity, the curved structure exhibits multiple stable equilibria, snap-through motion and pull-in instabilities along with modal interaction [18,19,20,21,22]. It is well acknowledged that there are abundant nonlinear phenomena in MEMS while extending the micro-devices driving forces into the nonlinear regimes [23,24,25,26,27,28,29,30,31]. Essentially, a large excitation will significantly increase the vibration amplitude and thus improve the performance of such devices [32,33,34].

However, the instability usually coexists with any large nonlinear behavior, and therefore, it is indispensable to explore how to increase the vibration amplitude without reducing the stability of the designed micro-sensor. Nevertheless, and despite the challenging concerns coming from the nonlinear behavior, electrostatically actuated MEMS resonators have been utilized as sensors mainly utilizing the resonance frequency shift to measure forces, accelerations and masses [20,25]. To this extent, a new generation of MEMS based sensors using the phenomenon of mode localization (confinement of vibration) has emerged in the past decade, demonstrating rich and complex interesting nonlinear modal interaction behaviors, such as mode crossing and mode veering [34]. This mainly happens when two frequencies of a system are equal in magnitude or become very close to each other, respectively.

Indeed, some research works [35,36,37,38,39,40,41,42] have carefully analyzed the nonlinear characteristics of single resonator based micro-sensors in order to depict such interesting nonlinear modal interaction behaviors [43]. Then, a few groups attempted to design coupled structures to exploit the mode coupling phenomenon and to design highly sensitive sensors [42,44,45]. They utilized the mode localization of mechanically and/or electrostatically coupled resonators operating in a linear vibration range and where mode aliasing represents an obstacle that limits their sensitivity improvement and therefore investigated mode localization to enhance the performance of these micro-sensors.

In this regard, the aim of the proposed work is to suggest and examine a novel MEMS sensor design consisting of two microbeams coupled and excited electrostatically via one lower in-plane stationary electrode. The design has the advantage of coupling electrostatically two microbeams serving as sensors to reduce the required input power and the device input power and footprint compared to that driven by comb-drive fingers. The two beams can be fabricated as straight and/or initially curved beams considering the advantages mentioned earlier in the literature review survey. The use of two different beam configurations in one device is beneficiary toward detecting two potential substances at the same time. The system would be a representative design of a mode-localized based sensor consisting of two weakly electrostatically coupled microbeams. A comprehensive analytical model incorporating all possible nonlinear terms is presented and solved based on the Galerkin method combined with the Long-Time Integration (LTI) method. The dynamic characteristics of two coupled resonators are investigated around their respective primary resonances.

## 2. Device Geometrical Properties and Operational Mechanism

The proposed sensor consists of two microbeams, both designed as straight or initially curved, denoted as upper and lower beams and actuated by one sidewall electrode, as shown in Figure 1. The two beams have a length of $\ell_{b}=1000$ µm and a width of b=30 µm. They both assume a thickness of h=2 µm whereas the initially curved beam assumes an initial rise of $b_{u\circ }=2\;$µm. The capacitor gap between the sidewall electrode to the straight beam is set to $d_{l}=10$ µm and from the straight beam to the initially curved beam is set to $d_{u}=10$ µm.

The sensor uses the phenomenon of mode localization and measures the change in the vibration mode of coupled resonators to detect a small perturbation. Its sensitivity depends on the ratio between the mechanical properties of the microbeams and the electrostatic coupling force. We expect that any small perturbation in the added mass will lead to veering and/or crossover phenomena, and therefore a change in the resonance frequency can be measured. The sensor’s sensitivity also depends on the operational mode where there are two fundamental modes. The first one is the quasi-static mode known in the literature as binary sensors with ON and OFF states. The second mode is based on a harmonic signal where the sensor can be operated as a traditional linear frequency-shift-based sensor or bifurcation-type based sensor depending on the actuation levels. We expect that the bifurcation-type sensors are significantly sensitive compared to the quasi-static and linear-frequency shift sensors to changes in mass interpreted as changes in the location of the bifurcation nodes in the nonlinear frequency responses.

The operational principle of the proposed design depends on the vibrating modes of the microbeams, as shown in the simple lumped model of Figure 2. These eventually correspond to the symmetric and the antisymmetric modes, respectively. This means that by adding a small mass on the first resonator and tuning the forcing static component, the fundamental frequency reduces, and therefore the coupled resonators sensor vibrates at the lowest mode and vise versa for the second resonator, as portrayed in Figure 2b. Indeed, this confirms that the fundamental resonating mode of the system becomes localized on the first resonator for the first mode, and on the second one for) the second mode. The sensitivity of such a sensor can be enhanced by tuning the material (stiffness coefficients $k_{1}$ and $k_{2}$ in the below Figure 2a), the geometrical (sizes of the vibrating masses) and the coupling electrostatic field.

## 3. Mathematical Modeling

The problem formulation and derivation of the equations of motion describing the transverse motion of the above sensor have been carried out following the Newton’s mechanics approach. The two beams are coupled electrostatically as shown in Figure 1. The initial curvature of the upper microbeam is denoted as $b_{u\circ }$. The respective cross-sectional area of the two beams is denoted by A=b∗h, respectively, where *h* is the thickness and *b* is the width. Knowing that both beams are anchored and as a first step, we draw the free-body-diagram (F.B.D) for each beam as shown in Figure 3 and Figure 4.

The forcing terms appearing in both two figures (${\hat{N}}_{l},{\hat{N}}_{u},{\hat{F}}_{ld},{\hat{F}}_{ud},{\hat{R}}_{l}\;\mathrm{and}\;{\hat{R}}_{u}$) are the normal, damping and mid-plane stretching forces for the lower and upper beams, respectively. ${\hat{F}}_{el}$ is the electrostatic force that actuates the lower microbeam from the stationary electrode. Note that the separation distance at any point along the lower beam to the stationary electrode is denoted by $\left(d_{l}-{\hat{w}}_{l}\right)$.

${\hat{F}}_{eu}$ is the electrostatic force applied to the upper beam and the separated distance between two points along the beam span is $\left(d_{u}+{\hat{w}}_{l}-{\hat{w}}_{u}\right)$. This, in fact, is where the two microbeams are coupled through the electrostatic force. It means that the force is directly applied to the lower beam and then the lower beam is excited the upper beam. The expressions of the forcing terms appearing in the F.B.Ds are all listed in the following Table 1:

Where $\hat{c}$ is the viscous damping coefficient, *E* is Young’s modulus of elasticity. $\ell_{b}$ is the length of the beam, $\hat{N}$ is the axial load, ε is the dielectric constant, $V_{dcl}$ and $V_{acl}$ are the applied biased and harmonic voltages in the lower beam. $V_{dcu}$ and $V_{acu}$ are the biased and time-varying voltages that apply to the upper beam. The excitation frequency of the voltage waveform is denoted by $\hat{\Omega }$.

### 3.1. Equations of Motion

Following the Newton’s second law associated with the Euler Bernoulli’s beam theory, we derive the equations of the motion describing the transverse deflections of the two microbeams as follows [8,10]:lower microbeam
(1)$$\begin{array}{cc} \rho A{\hat{\ddot{w}}}_{l}+\hat{c}{\hat{\dot{w}}}_{l}+EI{\hat{w}}_{l}^{⁗}= & \frac{EA}{2\ell_{b}}\left[{\hat{w}}_{l}^{\prime \prime }\int_{0}^{\ell_{b}}\left({{\hat{w}}_{l}^{\prime }}^{2}-2{\hat{w}}_{l}^{\prime }\right)\hat{x}\right]+\frac{\varepsilon bV_{l}^{2}}{2{\left(d_{l}-{\hat{w}}_{l}\right)}^{2}}\\ & -\frac{\varepsilon bV_{u}^{2}}{2{\left(d_{u}+{\hat{w}}_{l}+{\hat{w}}_{u\circ }-{\hat{w}}_{u}\right)}^{2}} \end{array}$$
where
$${\hat{w}}_{u\circ }=\frac{b_{u\circ }}{2d_{u}}\left(1-\cos\left(2\pi x\right)\right)$$
and the associated boundary conditions are
(2)$${\hat{w}}_{l}\left(0,\hat{t}\right)=0\;,\;{\hat{w}}_{l}^{\prime }\left(0,\hat{t}\right)=0\;,\;{\hat{w}}_{l}\left(\ell_{b},\hat{t}\right)=0\;,\;{\hat{w}}_{l}^{\prime }\left(\ell_{b},\hat{t}\right)=0$$

upper microbeam

(3)$$\begin{array}{cc} \rho A{\hat{\ddot{w}}}_{u}+\hat{c}{\hat{\dot{w}}}_{u}+EI{\hat{w}}_{u}^{⁗}= & \frac{EA}{2\ell_{b}}\left[\left({\hat{w}}_{u}^{\prime \prime }-{\hat{w}}_{u\circ }^{\prime \prime }\right)\int_{0}^{\ell_{b}}\left({{\hat{w}}_{u}^{\prime }}^{2}-2{\hat{w}}_{u\circ }^{\prime }{\hat{w}}_{u}^{\prime }\right)d\hat{x}\right]\\ & +\frac{\varepsilon bV_{u}^{2}}{2{\left(d_{u}+{\hat{w}}_{l}+{\hat{w}}_{u\circ }-{\hat{w}}_{u}\right)}^{2}} \end{array}$$
with associated boundary conditions listed as
(4)$${\hat{w}}_{u}\left(0,\hat{t}\right)=0\;,\;{\hat{w}}_{u}^{\prime }\left(0,\hat{t}\right)=0\;,\;{\hat{w}}_{u}\left(\ell_{b},\hat{t}\right)=0\;,\;{\hat{w}}_{u}^{\prime }\left(\ell_{b},\hat{t}\right)=0$$
where ρ denotes the mass density and *I* symbolizes the moment of inertia, which is equal to $\frac{bh^{3}}{12}$. Equation (Equation 1) shows that the two beams are electrostatically coupled through the force term. This, in fact, confirms that as the lower beam is excited, the upper beam will respond to it and may result in rich static and dynamic behavior depending on the excitation level. This configuration could also lead to several excitation scenarios as will be discussed in the following sections.

### 3.2. Normalization Process

In order to deal with these types of equations in the micro-scale, it is more convenient to write them in the nondimensional form. Thus, this process can be performed by introducing the following nondimensional variables:(5)$$w_{l}=\frac{{\hat{w}}_{l}}{d_{l}}\;,\;w_{u}=\frac{{\hat{w}}_{u}}{d_{u}}\;,\;x=\frac{\hat{x}}{\ell_{b}}\;,\;t=\frac{\hat{t}}{T}$$
where *T* is a time scale parameter and is chosen to be $T=\sqrt{\rho bh\ell_{b}^{4}/EI}$. Substituting Equation (Equation 5) into Equations (Equation 1)–(Equation 4) yield to the nondimensional equation of motion of

lower microbeam

(6)$$\begin{array}{cc} {\ddot{w}}_{l}+c{\dot{w}}_{l}+EIw_{l}^{⁗}= & \alpha_{1}\left[w_{l}^{\prime \prime }\int_{0}^{1}\left({w_{l}^{\prime }}^{2}-2w_{l}^{\prime }\right)dx\right]+\frac{\alpha_{2}V_{l}^{2}}{{\left(1-w_{l}\right)}^{2}}\\ & -\frac{\alpha_{2}V_{u}^{2}}{{\left(\frac{d_{u}}{d_{l}}+w_{l}+w_{u\circ }-w_{u}\right)}^{2}} \end{array}$$
and the associated boundary conditions are
(7)$$w_{l}\left(0,t\right)=0\;,\;w_{l}^{\prime }\left(0,t\right)=0\;,\;w_{l}\left(1,t\right)=0\;,\;w_{l}^{\prime }\left(1,t\right)=0$$

upper microbeam

(8)$$\begin{array}{cc} {\ddot{w}}_{u}+c{\dot{w}}_{u}+EIw_{u}^{⁗}= & \alpha_{1}\left(w_{u}^{\prime \prime }-w_{u\circ }^{\prime \prime }\right)\left[\int_{0}^{1}\left({w_{u}^{\prime }}^{2}-2w_{u\circ }^{\prime }w_{u}^{\prime }\right)dx\right]\\ & +\frac{\alpha_{2}V_{u}^{2}}{{\left(\frac{d_{u}}{d_{l}}+w_{l}+w_{u\circ }-w_{u}\right)}^{2}} \end{array}$$
with associated boundary conditions listed as
(9)$$w_{u}\left(0,t\right)=0\;,\;w_{u}^{\prime }\left(0,t\right)=0\;,\;w_{u}\left(1,t\right)=0\;,\;w_{u}^{\prime }\left(1,t\right)=0$$
where the nondimensional coefficients appear in Equations (Equation 6) and (Equation 8) are defined as follows
(10)$$c=\frac{\hat{c}\ell_{b}^{4}}{TEI}\;,\;\alpha_{1}=6{\left(\frac{d_{l}}{h}\right)}^{2}\;,\;\alpha_{2}=6\left(\frac{\varepsilon \ell_{b}^{4}}{Eh^{3}d_{l}^{3}}\right)$$

### 3.3. Reduced-Order Model (ROM)

Equations (Equation 6) and (Equation 8) are discretized using the straight beam mode shapes $\phi_{i}\left(x\right)$ as basis functions in a Galerkin expansion to obtain the reduced-order model (ROM). First of all, we solve for the static deflection of the both two beams $w_{ls}$ and $w_{us}$ as a function of the static voltages $V_{ldc}$ and $V_{udc}$ by eliminating the time derivatives from the equations of motion to obtain a static equilibrium equations. This results in a static equation for the lower microbeam as follows:(11)$$\begin{array}{cc} w_{ls}^{⁗}=\alpha_{1}\left[w_{ls}^{\prime \prime }\int_{0}^{1}\left(w_{ls}^{\prime \prime }-2w_{ls}^{\prime }\right)dx\right] & +\frac{\alpha_{2}V_{ldc}^{2}}{{\left(1-w_{ls}\right)}^{2}}-\frac{\alpha_{2}V_{udc}^{2}}{{\left(\frac{d_{u}}{d_{l}}+w_{ls}+w_{u\circ }-w_{us}\right)}^{2}} \end{array}$$
and it is subjected to the following boundary conditions
(12)$$w_{ls}\left(0,t\right)=0\;,\;w_{ls}^{\prime }\left(0,t\right)=0\;,\;w_{ls}\left(1,t\right)=0\;,\;w_{ls}^{\prime }\left(1,t\right)=0$$

Similarly, the equation describing the static equilibria of the upper beam under the effect of the static voltage can be written as:(13)$$\begin{array}{c} w_{us}^{⁗}=\alpha_{1}\left(w_{us}^{\prime \prime }-w_{u\circ }^{\prime \prime }\right)\left[\int_{0}^{1}\left(w_{us}^{\prime \prime }-2w_{u\circ }^{\prime }w_{us}^{\prime }\right)dx\right]+\frac{\alpha_{2}V_{udc}^{2}}{{\left(\frac{d_{u}}{d_{l}}+w_{ls}+w_{u\circ }-w_{us}\right)}^{2}} \end{array}$$
and it is subjected to the following boundary conditions
(14)$$w_{us}\left(0,t\right)=0\;,\;w_{us}^{\prime }\left(0,t\right)=0\;,\;w_{us}\left(1,t\right)=0\;,\;w_{us}^{\prime }\left(1,t\right)=0$$

Then, we descretize the static deflections of the two microbeams in terms of the Galerkin approximation as
(15)$$\begin{array}{cc} w_{ls} & =\sum_{i=1}^{N}\phi_{i}\left(x\right)q_{li}\;;\;i=1,\ldots ,N\\ w_{us} & =\sum_{i=1}^{N}\phi_{i}\left(x\right)q_{ui}\;;\;i=1,\ldots ,N \end{array}$$
where $q_{li}$ are modal coordinates for the lower beam and $q_{ui}$ are modal coordinates for the upper beam. Substituting these transformation forms into Equations (Equation 11)–(Equation 14) and then multiplying both sides of Equation (Equation 11) by $\left[{\left(1-w_{ls}\right)}^{2}\times {\left(\frac{d_{u}}{d_{l}}-w_{ls}-w_{u\circ }-w_{us}\right)}^{2}\right]$ to avoid numerical errors in the response near the singularity yield to the following ROM for the lower microbeam as
(16)$$\begin{array}{cc} \; & \left({\left(1-\sum_{i=1}^{N}\phi_{i}q_{li}\right)}^{2}{\left(\frac{d_{u}}{d_{l}}-\sum_{i=1}^{N}\phi_{i}q_{li}-w_{u\circ }-\sum_{i=1}^{N}\phi_{i}q_{ui}\right)}^{2}\right)\;(\sum_{i=1}^{N}\phi_{i}^{iv}q_{li}\\ & -\alpha_{1}\left(\sum_{i=1}^{N}\phi_{i}^{{}^{\prime \prime }}q_{li}\right)\int_{0}^{1}\left({\left(\sum_{i=1}^{N}\phi_{i}^{{}^{\prime }}q_{li}\right)}^{2}-2\sum_{i=1}^{N}\phi_{i}^{{}^{\prime }}q_{li}\right)dx)+\frac{\alpha_{2}V_{ldc}^{2}}{{\left(1-\sum_{i=1}^{N}\phi_{i}^{{}^{\prime \prime }}q_{li}\right)}^{2}}\\ & -\frac{\alpha_{2}V_{udc}^{2}}{{\left(\frac{d_{u}}{d_{l}}+\sum_{i=1}^{N}\phi_{i}^{{}^{\prime \prime }}q_{li}+w_{u\circ }-\sum_{i=1}^{N}\phi_{i}^{{}^{\prime \prime }}q_{ui}\right)}^{2}}=0 \end{array}$$
and for the upper microbeam as
(17)$$\begin{array}{cc} \; & {\left(\frac{d_{u}}{d_{l}}-\sum_{i=1}^{N}\phi_{i}q_{li}-w_{u\circ }-\sum_{i=1}^{N}\phi_{i}q_{ui}\right)}^{2}\;(\sum_{i=1}^{N}\phi_{i}^{iv}q_{ui}-\alpha_{1}\left(\sum_{i=1}^{N}\phi_{i}^{{}^{\prime \prime }}q_{ui}-w_{u\circ }^{\prime \prime }\right)\int_{0}^{1}({\left(\sum_{i=1}^{N}\phi_{i}^{{}^{\prime }}q_{ui}\right)}^{2}\\ & -2w_{u\circ }^{\prime }\sum_{i=1}^{N}\phi_{i}^{{}^{\prime }}q_{ui})dx)+\frac{\alpha_{2}V_{udc}^{2}}{{\left(\frac{d_{u}}{d_{l}}+\sum_{i=1}^{N}\phi_{i}^{{}^{\prime \prime }}q_{li}+w_{u\circ }-\sum_{i=1}^{N}\phi_{i}^{{}^{\prime \prime }}q_{ui}\right)}^{2}}=0 \end{array}$$

Multiplying Equations (Equation 16) and (Equation 17) by the mode shapes $\phi_{j}$ and carrying out the integration over the beam length results in *N* algebraic equations describing the equilibrium position for the lower beam as
(18)$$\begin{array}{cc} \; & \int_{0}^{1}\phi_{j}[\left({\left(1-\sum_{i=1}^{N}\phi_{i}q_{li}\right)}^{2}{\left(\frac{d_{u}}{d_{l}}-\sum_{i=1}^{N}\phi_{i}q_{li}-w_{u\circ }-\sum_{i=1}^{N}\phi_{i}q_{ui}\right)}^{2}\right)\;(\sum_{i=1}^{N}\phi_{i}^{iv}q_{li}\\ & -\alpha_{1}\left(\sum_{i=1}^{N}\phi_{i}^{{}^{\prime \prime }}q_{li}\right)\int_{0}^{1}\left({\left(\sum_{i=1}^{N}\phi_{i}^{{}^{\prime }}q_{li}\right)}^{2}-2\sum_{i=1}^{N}\phi_{i}^{{}^{\prime }}q_{li}\right)dx)+\frac{\alpha_{2}V_{ldc}^{2}}{{\left(1-\sum_{i=1}^{N}\phi_{i}^{{}^{\prime \prime }}q_{li}\right)}^{2}}\\ & -\frac{\alpha_{2}V_{udc}^{2}}{{\left(\frac{d_{u}}{d_{l}}+\sum_{i=1}^{N}\phi_{i}^{{}^{\prime \prime }}q_{li}+w_{u\circ }-\sum_{i=1}^{N}\phi_{i}^{{}^{\prime \prime }}q_{ui}\right)}^{2}}]=0 \end{array}$$
and for the upper beam as
(19)$$\begin{array}{cc} \; & \int_{0}^{1}\phi_{j}[{\left(\frac{d_{u}}{d_{l}}-\sum_{i=1}^{N}\phi_{i}q_{li}-w_{u\circ }-\sum_{i=1}^{N}\phi_{i}q_{ui}\right)}^{2}\;(\sum_{i=1}^{N}\phi_{i}^{iv}q_{ui}-\alpha_{1}\left(\sum_{i=1}^{N}\phi_{i}^{{}^{\prime \prime }}q_{ui}-w_{u\circ }^{\prime \prime }\right)\\ & \int_{0}^{1}\left({\left(\sum_{i=1}^{N}\phi_{i}^{{}^{\prime }}q_{ui}\right)}^{2}-2w_{u\circ }^{\prime }\sum_{i=1}^{N}\phi_{i}^{{}^{\prime }}q_{ui}\right)dx)+\frac{\alpha_{2}V_{udc}^{2}}{{\left(\frac{d_{u}}{d_{l}}+\sum_{i=1}^{N}\phi_{i}^{{}^{\prime \prime }}q_{li}+w_{u\circ }-\sum_{i=1}^{N}\phi_{i}^{{}^{\prime \prime }}q_{ui}\right)}^{2}}]=0 \end{array}$$

These equations are then solved for $q_{li}$ and $q_{ui}$ as functions of the static voltage to obtain the static deflections of the lower and upper microbeams. The forced eigenvalue problem describing the two microbeams oscillations around their static equilibria is obtained by resolving each beam total deflection into a static component and a dynamic component and then substituting this form into the equations of motion. Then, a similar procedure to that used in the static analysis can be carried out to develop the reduced-order model of the eigenvalue problem. The resulting equations are then evaluated to obtain the dynamic responses of the two beams. This can be completed by integrating the equations over a long-time period and the time histories are then evaluated over the last signal periods to obtain the steady-state response.

## 4. Results and Discussions

In this investigation, two main actuation scenarios will be considered. The first scenario is to vary the lower beam’s actuation signal and keep it at a constant level for the upper beam. Then, we will revert the process by applying a varying signal to the upper beam and keep the lower beam actuation signal at a constant value. This is a required step to study the nonlinear dynamics and modal interactions of the two beams and how the variation in the control parameter would affect their static, eigenvalues and dynamic responses.

### 4.1. Case I: Varying Excitation Signal of the Lower Beam

Here, we investigate the static and dynamic responses of the coupled beams when the excitation signal varies along the lower beam and the upper beam $V_{udc}$ voltage’s sets to 0, 20 and 40 V, respectively. First of all, the static deflections of the lower and upper microbeams mid points $w_{ls}\left(0.5\right)$ and $w_{us}\left(0.5\right)$, respectively, excited by a distributed electrostatic force has been computed through simultaneously solving Equations (Equation 18) and (Equation 19) utilizing three symmetric mode in the Galerkin’s expansion. Figure 5a indicates that only the lower beam is responding to the actuation signal when the upper beam voltage is set to zero. The figure also shows that the lower beam mid-point deflection increases as the voltage increases until it reaches a pull-in voltage and loses its stability.

Similar behavior is also observed when the static voltage of the upper beam increases to 20 V. However, its mid-point deflection starts changing from a new setting-point corresponding to $w_{us}=0.145$ µm as shown in Figure 5b. This is expected because the upper beam in this case is biased. On the other hand, the lower beam mid-point deflection starts increasing its amplitude from a setting point of $w_{ls}=-0.326$ µm. We note that setting the actuation dc voltage of the upper microbeam to 20 V or less is not sufficient enough to actuate it. Indeed, we confirm there will be no modal interaction between the two beams for this particular case.

To further validate this, we increase the upper microbeam voltage to 40 V. This leads to a highly nonlinear static response characterized by a snap through where the stable branches of the solution meet the unstable branches of the solution, as shown in Figure 5c. The stable branches of solutions are marked as solid lines and the unstable branches of the solution are marked as dashed lines. The results show also that the upper beam is now activated and responding to the actuation signal. We believe this is due to a high electric field with energy streaming or flowing from the lower beam to the upper beam.

Next, we explore the effect of these excitation signals on the fundamental frequencies of both microbeams. This is performed by substituting the static results obtained above, employing a ROM with three symmetric modes into the equation described in Section 3.3 and then solving for the corresponding eigenvalues. Because of the presence of the coupled electrostatic forces between the two beams, we expect some nonlinear phenomena, such as mode veering, mode crossover and eventually mode localization.

We note that the veering phenomenon may occur when one of the eigenvalues of the two beams approach each other as the excitation signal varies. This is a similar behavior to that presented in [21,22,34,39]. Then, they diverge away from each other as the control parameter leaves the zone. These frequencies’ closeness could increase the sensitivity of such a design. To confirm it, we have to track the changes in the resonance frequencies of the two microbeams. At this stage, we are targeting the frequencies that are corresponding to the first in-plane symmetric mode.

Figure 6 shows the variation in the first resonance frequency of the lower beam $f_{l1}$, marked with blue lines, and of the upper beam $f_{u1}$, marked with orange lines, as the lower beam static voltage varies and the upper beam’s voltage is set to 0, 20 V and 40 V, respectively. We note that the upper beam resonance frequency does not display any change, as shown in Figure 6a,b. This is expected because the static deflection of the upper beam $w_{us}$ does not change due to the weak electrostatic coupling force. The figure also shows that a crossover phenomenon occurs between the two frequencies.

Increasing the upper static voltage to $V_{udc}=40$ V, the two resonance frequencies, $f_{l1}$ and $f_{u1}$, approach each other as the lower dc voltage reaches 40 V, as illustrated in Figure 6c. At this zone, the lower beam frequency increases and the upper beam frequency decreases. This closeness is characterized by a veering phenomenon with a minimum difference between the two frequencies of Δf=2.63 kHz and it occurs at a voltage of $V_{ldc}=45$ V.

Furthermore, we examine the frequency-response curves of the first lower ($f_{l1}$) and upper ($f_{u1}$) resonance frequencies in the vicinity of the veering zone shown in Figure 6c. This is completed by subjecting the sensor to three different voltage waveforms. They correspond to voltages before, at and after the veering zone. Utilizing different excitation voltages will give further insights about the dynamic interaction between the two microbeams and how the energy is exchanging among them. In this study, the quality factor was set to 100, the harmonic voltage to $V_{lac}=0.1$ V and the upper static voltage to $V_{udc}=40$ V.

Just before the veering zone, the frequency-response curves of the two beams show linear responses with a mild softening behavior when the lower static voltage sets to $V_{ldc}=40$ V, as shown in Figure 7a. The frequency gap between the two resonances is approximately found to be Δf=4.55 kHz. We note that the dynamic oscillations of the lower beam are much higher than that of the upper beam and thus it becomes more sensitive. This, in fact, is due to the effect of the electrostatic force, which is localized at the lower mode compared to that at the higher mode. The figure also shows a modal interaction characterized by two small peaks appearing at each resonance. This means the two microbeams start to exchange energy as they move forward and closer to the veering zone.

To further examine the modal interaction, we increase the lower static voltage to $V_{ldc}=45$ V. This value is even closer to the veering zone. Figure 7b indicates that the two resonances become close to each other with a frequency gap of Δf=2.63 kHz. As the excitation frequency is swept up, two additional peaks appear at the lower and the upper microbeams’ resonances, respectively. This confirms that the two beams are electrostatically coupled and the energy is equally distributed across each mode. This behavior is suitable for mass sensing applications because of the presence of two microbeams that allow detecting two different objects at the same time while being excited by a single force.

On the other hand, Figure 7c shows that the two modes veer away from each other as the lower static voltage increases to $V_{ldc}=50$ V with a frequency gap of Δf=3.11 kHz. This confirms that the two modes are leaving the veering zone with no sign of crossing. We note that the energy is localized at the upper beam. However, the lower mode dominates the dynamic oscillation in the neighborhood of the upper mode.

We can conclude that before the veering zone, each mode is stronger in its region with a weak electrostatic coupling force. This is not the case as they move toward the veering zone where both of them are equally contributing to the overall dynamic response, after which the energy is localized at the vicinity of the upper beam with a high contribution coming from the lower mode.

### 4.2. Case II: Varying Excitation Signal of the Upper Beam

In the subsequent simulations, we apply a varying excitation waveform along the upper beam and set the lower beam to constant values corresponding to 0, 20 and 40 V. Figure 8 shows the variation in the mid-point static deflection of the lower $w_{ls}\left(0.5\right)$ and of the upper beam $w_{us}\left(0.5\right)$ as a function the upper beam excitation voltage. In all cases, two branches of stable equilibria marked as solid lines and two branches of unstable equilibria marked as dashed black lines were observed.

Varying the upper beam voltage only shows that the mid-point deflection of the lower beam decreases as the upper voltage increases along the first branch of stable equilibria, corresponding to the beam initial curvature. Similarly, the upper beam deflection increases until $V_{udc}$ voltage reaches 39.33 V, where the two beams jump to a second equilibrium corresponding to the initial counter-curvature at point marked as ST. This jump is a basic characteristic of the snap-through process. At this point, the stable branches of equilibria meet the first branch of unstable equilibria in a saddle-node bifurcation as shown in Figure 8a. Under this actuation level, we note that the two beams start deflecting from a zero position as clearly shown in the figure.

On the other hand, increasing the upper static voltage beyond the snap-through threshold increases the counter deflection of the upper mid point and decreases the counter deflection of the lower mid point along the second stable branch until it reaches another saddle-node bifurcation denoting the “pull-in instability” at a $V_{udc}$ voltage of 42.67 V. At this point, the second branches of stable equilibria meet the second branches of unstable equilibria and lose their stability by going into contact with the stationary electrode. Indeed, there are no physical stable equilibria beyond this point.

Additionally, decreasing the upper voltage after the two beams snap toward the second equilibria decreases the counter curvature of the upper beam mid point and increases the counter curvature of the lower beam mid point along the second stable branches until the beams snap back (SB) and jump at V=38.27 V to the first branches of stable equilibria as illustrated in Figure 8a. At this point, the second stable branches of equilibria meet the first unstable branches in another saddle-node bifurcation.

A similar behavior is also observed when the lower voltage sets to $V_{ldc}=20$ V, as shown in Figure 8b. However, the lower beam starts deflecting from a biased position corresponding to $w_{ls}=0.42\;$µm. This, in fact, is due to the effect of the static voltage that applies directly to the lower beam before the upper voltage is activated. The results show that the deflections of the two beams are equal in magnitude as the upper voltage reaches $V_{udc}=19.36$ V. Repeating the similar procedure described above and further increasing the lower static voltage to $V_{ldc}=40$ V lead to a similar response, as shown in Figure 8c.

We note that, under this actuation scheme, the lower beam starts its deflection from a new biased position corresponding to $w_{ls}=1.49\;$µm and its first stable branch of solution crosses the upper first stable branch of solution at a voltage of $V_{ldc}=38.29$ V. This is closer to the snap-through region and confirms that the more the lower voltage increases the more the crossing point shifts to the right.

Moreover, the variation of the fundamental frequencies of the both microbeams under excitation waveforms similar to that used in the static analysis has been studied. This will give more insight as to whether the modes are veering or crossing each other under these excitation conditions. Setting the lower static voltage to $V_{ldc}=0$ and varying the upper static voltage lead to continuous increases in the lower beam fundamental frequency, marked with a blue line, before it drops along the first stable branch of equilibria until it reaches zero at the snap-through voltage $V_{udc}=39.33$ V, as shown Figure 9a.

Then, it increases as the upper voltage further increases along the second branch of equilibria, corresponding to the initial counter curvature, until it reaches a maximum value of 14.88 kHz. After that, it suddenly drops and reaches zero at the pull-in instability voltage of V=42.67 V. We note that the first drop indicates that the geometric nonlinearities dominate the electrostatic force nonlinearities. However, after the beam snaps, the beam becomes closer to the stationary electrode, and therefore the electrostatic force nonlinearities dominate the geometric nonlinearities and overcome its restoring force.

On the other hand, the upper beam fundamental frequency continuously drops along the first stable equilibria, marked with orange line, as the voltage increases until it reaches a zone where its value become closer to the lower beam fundamental frequency. This is a classical behavior of the veering phenomenon where two modes are approaching each other and then veer away as the control parameters change. In the neighborhood of veering zone, we have found that the minimum frequency gap between the two frequencies is approximately Δf=2.53 kHz and occurs at a voltage of $V_{udc}=31.3$ V.

Figure 9a illustrates that the upper beam frequency increases as it leaves the veering zone until it reaches the snap-through threshold. We note that the mode does not evince a discontinuity as the equilibrium position jumps from the first to the second stable branches of the solution at the snap-through point. However, it jumps to higher values after the snap through and does not reach zero neither at the snap through nor at the pull-in points due to their strong geometric nonlinearities as compared to the forcing nonlinearities.

A similar behavior is also observed when the lower static voltage increases to $V_{ldc}=20$ V. However, the two modes are approaching each other to a zone closer to the snap-through threshold compared to that when it sets to 0 V. The minimum gap between the two frequencies was found to be around 2.5 kHz, as shown in Figure 9b. Alternatively, the veering phenomenon disappears as the lower static voltage further increases to $V_{ldc}=40$ V, as clearly shown in Figure 9c. It confirms that not all coupled resonators would generate a veering zone. It totally depends on the design parameters as well as the excitation condition.

The dynamic response under this excitation scenario has been investigated by subjecting the two beams to a frequency sweep test in the vicinity of the veering zone, as described in Figure 9. In this analysis, the signal frequency was swept up in a frequency range of 10–30 kHz and the lower beam forcing signal was set to 0 and the upper beam forcing signal was set to three levels: before, at and after veering zone.

The variation of the two beam mid-point velocities obtained under a voltage waveform with $V_{ldc}=0$ V, $V_{udc}=20$ V and $V_{uac}=0.1$ V, corresponding to a voltage signal before the veering zone shown in Figure 9a, and a frequency swept up in the range of 10–30 kHz is shown in Figure 10a. It shows that as the signal frequency increases, the response increases until it hits the primary resonance of the lower beam marked with a blue line. Then, the response decreases as the frequency further increases until it reaches the vicinity of the upper beam primary resonance, marked with an orange line. Small peaks appear at each primary resonance due to the effect of the electrostatic coupling force between the two modes. We observed that the two resonances are far away from each other with a frequency gap of Δf=7.3 kHz.

Further increasing the upper static voltage leads to a fast attraction toward the veering zone with frequency gap starting to decrease, as shown in Figure 10b–d. During this examination, we found that the small peak appears in the vicinity of the lower beam and primary resonance grows dramatically as the forcing level increases. This confirms that as the two modes are penetrating, the veering zone with the upper beam resonance frequency dominates the lower resonance frequency. However, the two modes are not completely merged and produce a single peak because there is still a frequency gap, as shown in Figure 10d.

As the excitation force leaves the veering zone, the two modes start growing with energy showing a mode localization like in the vicinity of the lower beam primary resonance due to its shortcoming as shown in Figure 11a,b. It shows that the contribution of the upper beam is more dominant than that corresponding to the lower one. In addition, the frequency gap starts increasing, confirming that the two modes are moving away from each other. We also note that when the lower voltage sets to zero, the two frequency-response curves show softening behavior.

Replication of the above analysis with the lower voltage set to $V_{ldc}=20$ V results in a similar behavior. However, a hardening response was observed in the frequency-response curve of the lower beam whereas a softening response was observed for the upper beam before the veering zone, as shown in Figure 12. Both frequency-response curves have a discontinuity due to cyclic-fold bifurcation where the amplitude jumps from the lower oscillation to the upper oscillation regime. In fact, these mixed nonlinear responses switch to linear as the two resonances approach the veering zone. Finally, both resonances become softer as they leave the veering zone and move away from each other, as illustrated in Figure 13.

## 5. Conclusions

This investigation has explored the vibration of weakly electrostatically coupled clamped-clamped straight and initially curved microbeams. Linear (veering and crossing) and nonlinear (localization) modal interactions under primary resonance were examined for potential use of the design as a mode-localized based sensor. A comprehensive analytical model incorporating all nonlinearities was established and then numerically solved based on the Galerkin modal expansion method superimposed to a long-time integration technique. The dynamic characteristics of the two coupled resonators under primary resonance were generated, and the contributions of their respective primary resonances in the overall response were analyzed accordingly. The static results under dc voltages only showed possibilities of having a mix of one-stable and even two-stables operating conditions, both adjustable through slightly perturbing the assumed lower stationary electrode dc voltage. Subsequently, this control parameter showed as well, throughout an eigenvalue problem analysis, that for the coupled resonators, mode veering can occur, offering potentials of nonlinear modal interaction, such as mode localization. Then, a dynamic analysis was carried out and showed that under the same excitation conditions and around the system primary resonance, linear and even softening behaviors can be obtained by adjusting the overall actuating dc and/or ac voltage amplitudes. In addition, when assuming a dc voltage at the lower electrode that triggers possible modes veering, it was noticed that as this excitation force drives the coupled system in the neighborhood of such a veering zone, the two primary resonant modes are at maximum amplitudes, showing a non-trivial frequency gap. They start varying and show a mode localization like in the vicinity of the upper and/or lower beam primary resonance. This confirmed that as the two resonant peaks are penetrating the veering zone, the upper and lower microbeams’ respective frequencies gap reduces and they sort of exchange the peak-like dominance accordingly. Such interesting modal interactions could lead to an increase in the sensitivity of such a sensing mechanism and the computed results showed that this can be achieved by just varying a simple control dc voltage parameter. The use of two different beam configurations in one device showed the possibility of detecting two potential substances at the same time with two interacting distinct resonant peaks.

## Figures and Tables

**Figure 1 micromachines-13-00989-f001:**
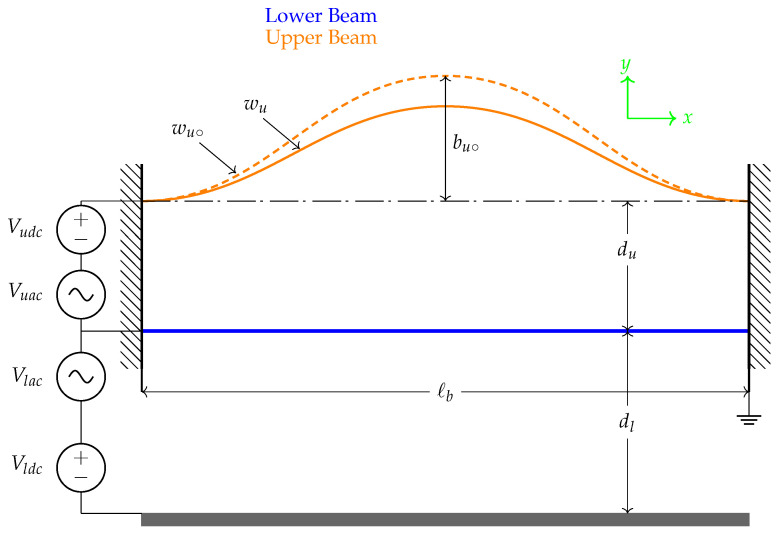
A schematic showing the sensor and the electrical connection.

**Figure 2 micromachines-13-00989-f002:**
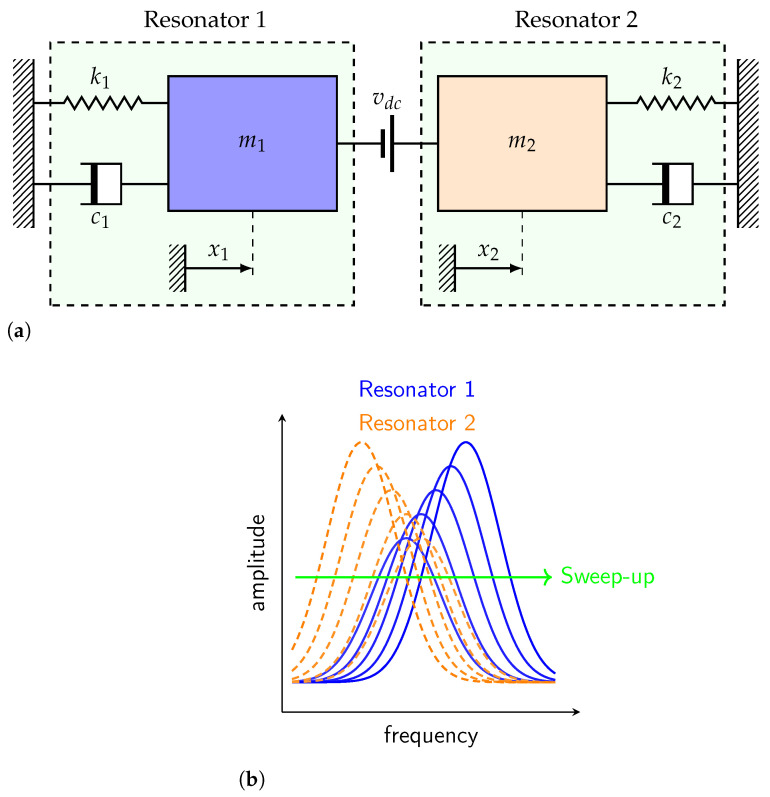
(**a**) A schematic showing the electrostatically coupled microbeams and (**b**) typical mode-localized based sensors frequency-response curves.

**Figure 3 micromachines-13-00989-f003:**
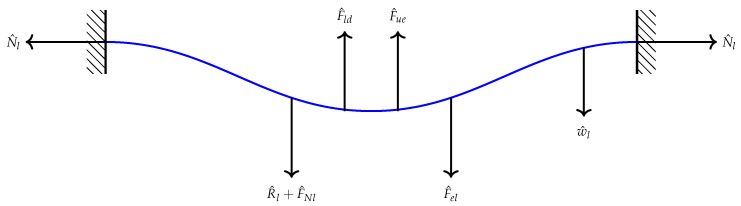
The F.B.D for the lower microbeam under the effect of electrostatic force, damping force, normal force and mid-pane stretching.

**Figure 4 micromachines-13-00989-f004:**
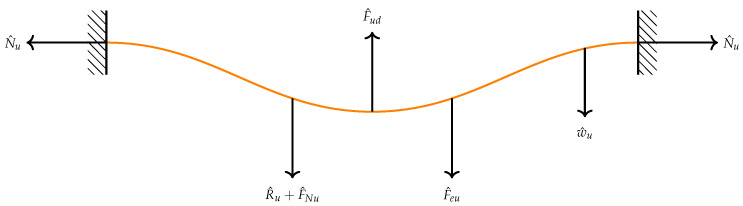
The F.B.D for the upper microbeam under the effect of electrostatic force, damping force, normal force and mid-pane stretching.

**Figure 5 micromachines-13-00989-f005:**
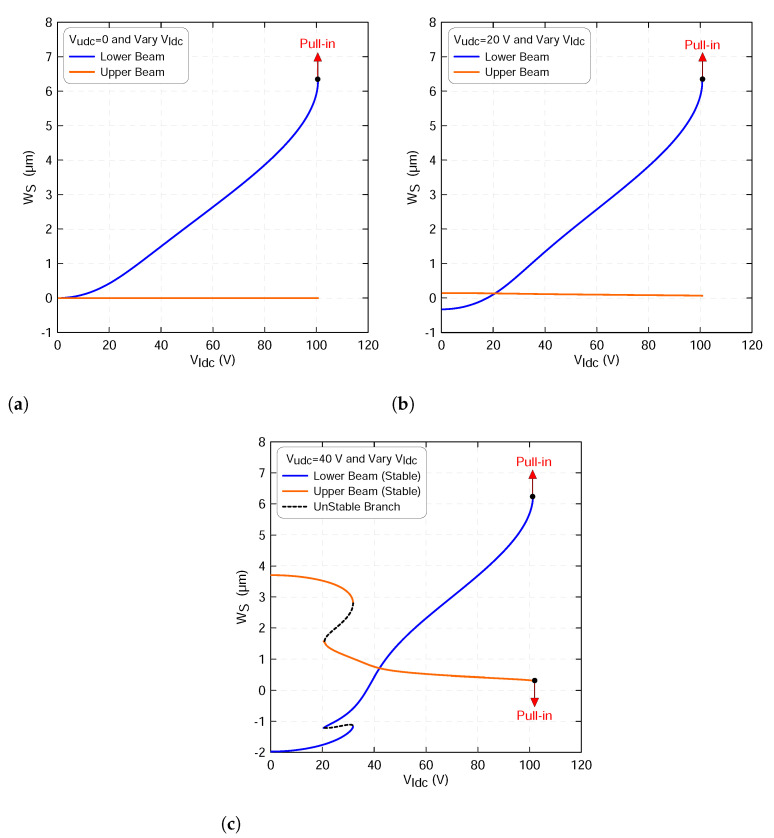
The beam mid-point deflections of the lower beam $w_{ls}\left(0.5\right)$ and of the upper beam $w_{us}\left(0.5\right)$ as a function of the lower static voltage using three-symmetric modes ROM and an upper voltage sets to: (**a**) $V_{udc}=0$, (**b**) $V_{udc}=20$ V and (**c**) $V_{udc}=40$ V. The stable static equilibria of the lower beam are marked with blue lines, the stable static equilibria of the upper beam are marked with orange lines whereas the unstable static equilibria are marked with dashed black lines.

**Figure 6 micromachines-13-00989-f006:**
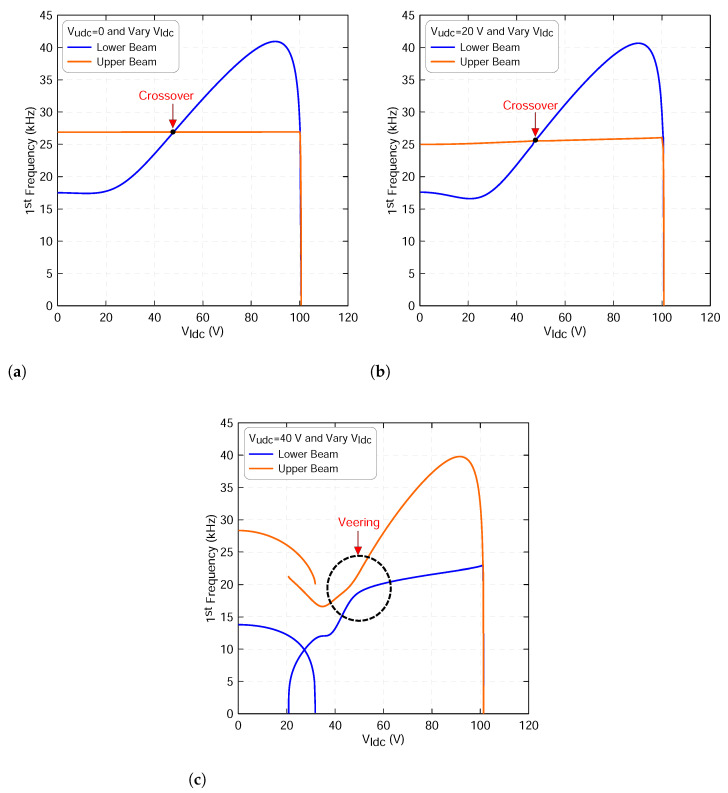
The variation in the first resonance frequency of the lower beam $f_{l1}$ and of the upper beam $f_{u1}$ as a function of the lower static voltage using three-symmetric modes ROM and an upper voltage sets to: (**a**) $V_{udc}=0$, (**b**) $V_{udc}=20$ V and (**c**) $V_{udc}=40$ V. The lower beam results are marked with blue lines and the upper beam results are marked with orange lines.

**Figure 7 micromachines-13-00989-f007:**
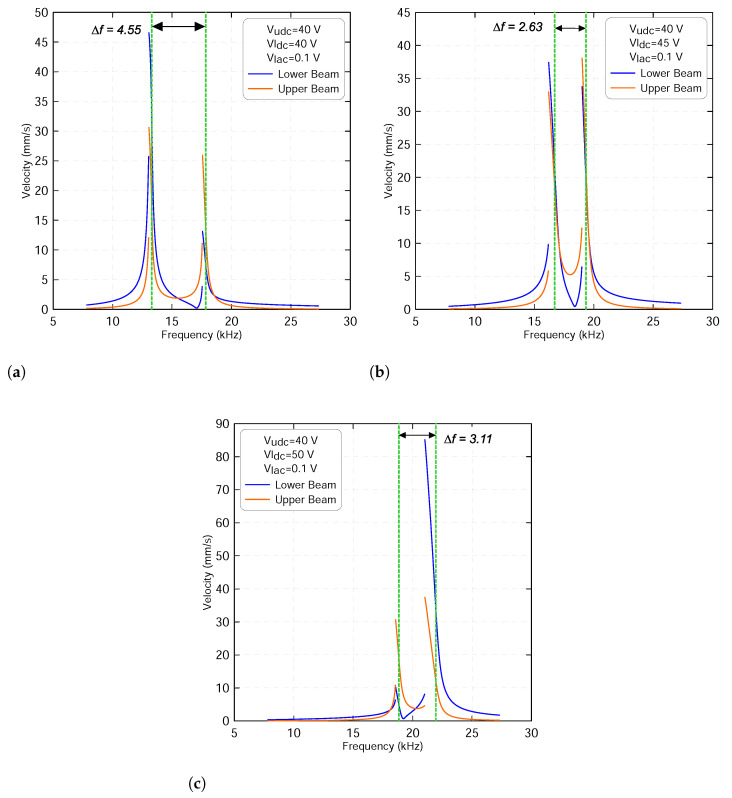
The frequency-response curves of the lower beam resonance frequency $f_{l1}$ and of the upper beam resonance frequency $f_{u1}$ with an upper voltage sets to $V_{udc}=40$ V and lower voltage sets to: (**a**) $V_{ldc}=40$ V, (**b**) $V_{ldc}=45$ V and (**c**) $V_{ldc}=50$ V. The lower harmonic voltage $V_{lac}$ is set to 0.1 V. The lower beam results are marked with blue lines and the upper beam results are marked with orange lines.

**Figure 8 micromachines-13-00989-f008:**
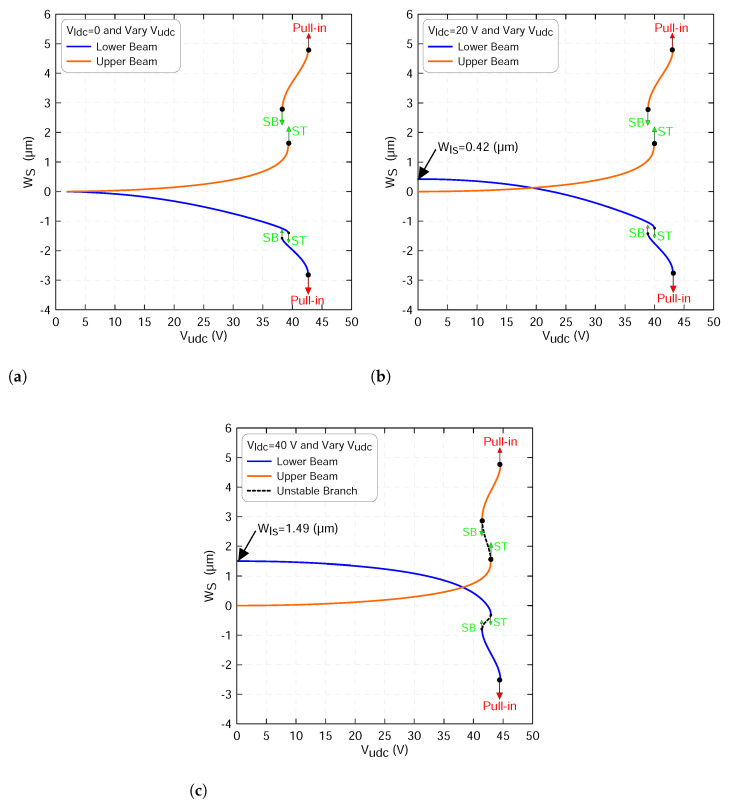
The beam mid-point deflections of the lower beam $w_{ls}\left(0.5\right)$ and of the upper beam $w_{us}\left(0.5\right)$ as a function of the upper static voltage using three-symmetric modes ROM and a lower voltage sets to: (**a**) $V_{ldc}=0$, (**b**) $V_{ldc}=20$ V and (**c**) $V_{ldc}=40$ V. The stable static equilibria of the lower beam are marked with blue lines, the stable static equilibria of the upper beam are marked with orange lines whereas the unstable static equilibria are marked with dashed black lines.

**Figure 9 micromachines-13-00989-f009:**
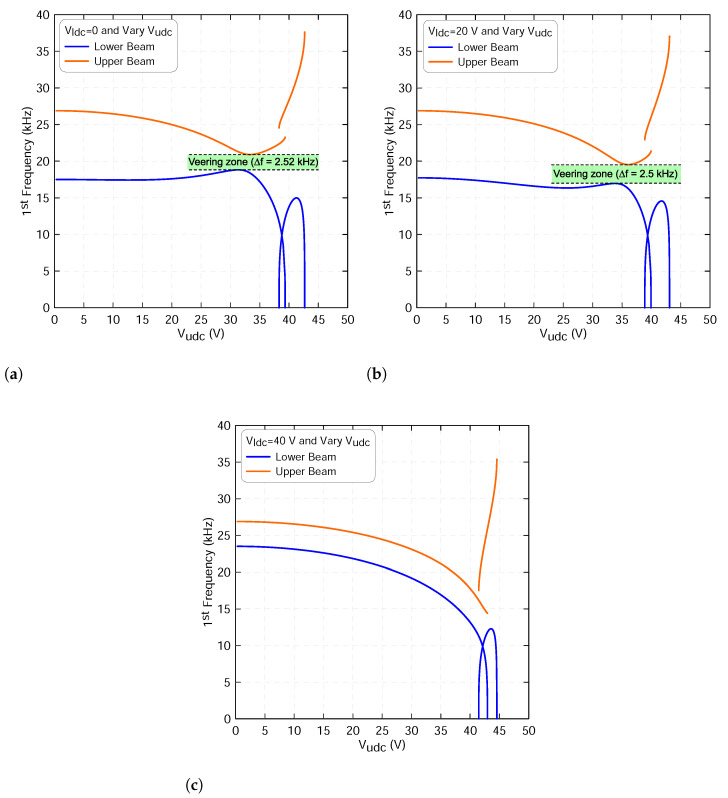
The variation in the first resonance frequency of the lower beam $f_{l1}$ and of the upper beam $f_{u1}$ as a function of the upper static voltage using three-symmetric modes ROM and a lower voltage sets to: (**a**) $V_{ldc}=0$, (**b**) $V_{ldc}=20$ V and (**c**) $V_{ldc}=40$ V. The lower beam results are marked with blue lines and the upper beam results are marked with orange lines.

**Figure 10 micromachines-13-00989-f010:**
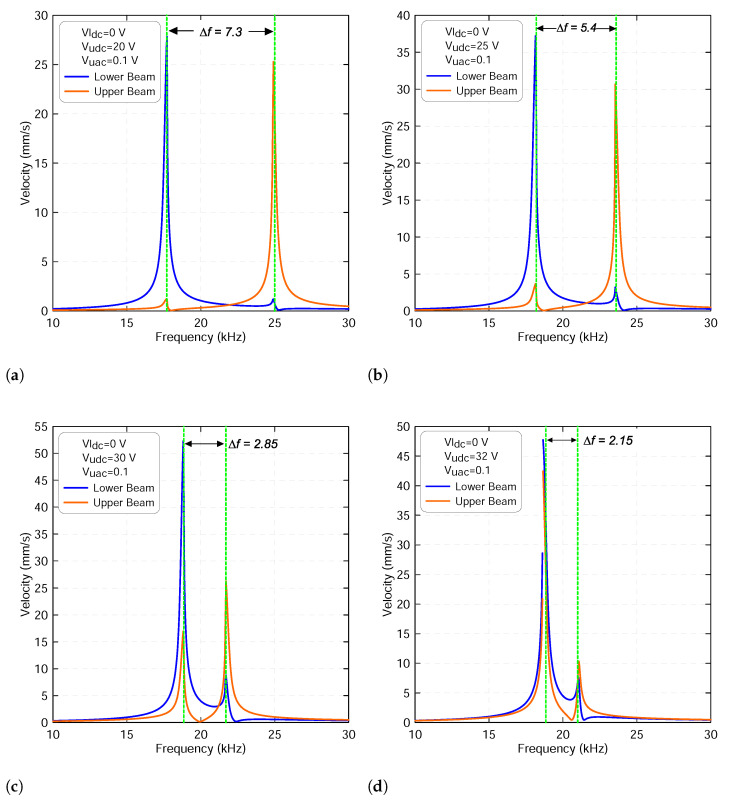
The frequency-response curves of the lower beam resonance frequency $f_{l1}$ and of the upper beam resonance frequency $f_{u1}$ with a lower voltage set to $V_{ldc}=0$ V and an upper voltage set to: (**a**) $V_{udc}=20$ V, (**b**) $V_{udc}=25$ V, (**c**) $V_{udc}=30$ V and (**d**) $V_{udc}=32$ V. The upper harmonic voltage $V_{uac}$ is set to 0.1 V. The lower beam results are marked with blue lines and the upper beam results are marked with orange lines.

**Figure 11 micromachines-13-00989-f011:**
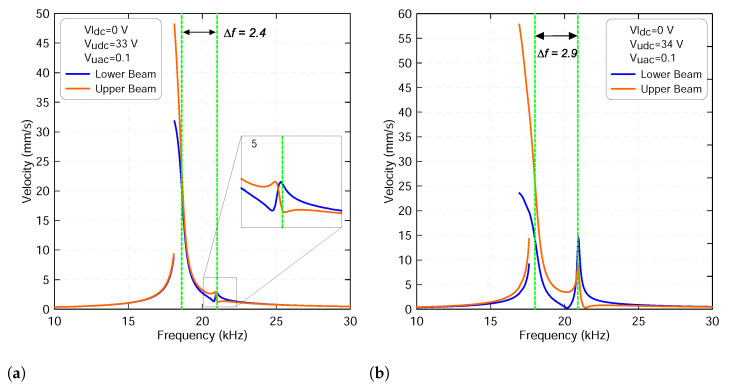
The frequency-response curves of the lower beam resonance frequency $f_{l1}$ and of the upper beam resonance frequency $f_{u1}$ with a lower voltage sets to $V_{ldc}=0$ V and a upper voltage sets to: (**a**) $V_{udc}=33$ V and (**b**) $V_{udc}=34$ V. The upper harmonic voltage $V_{uac}$ is set to 0.1 V. The lower beam results are marked with blue lines and the upper beam results are marked with orange lines.

**Figure 12 micromachines-13-00989-f012:**
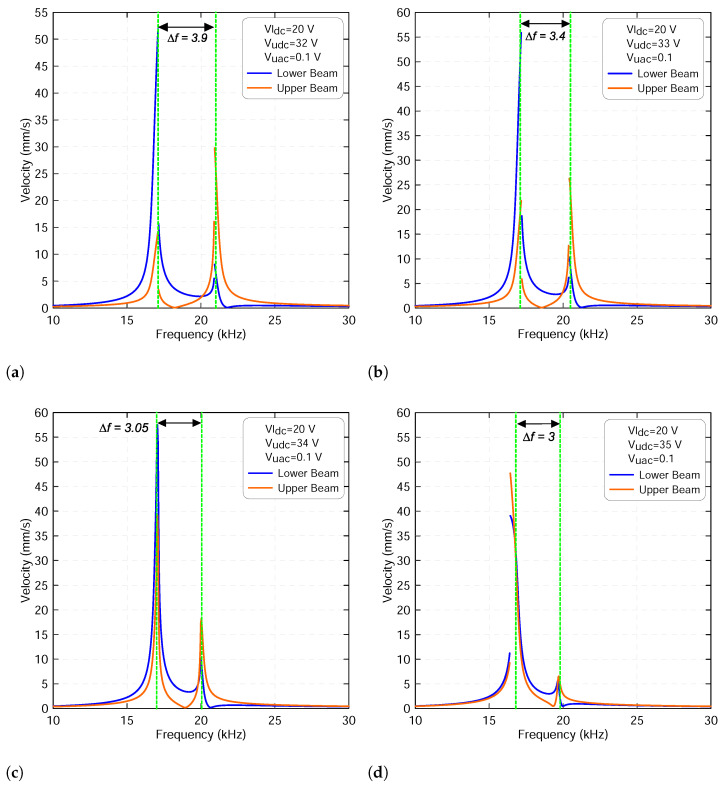
The frequency-response curves of the lower beam resonance frequency $f_{l1}$ and of the upper beam resonance frequency $f_{u1}$ with a lower voltage sets to $V_{ldc}=20$ V and a upper voltage sets to: (**a**) $V_{udc}=32$ V, (**b**) $V_{udc}=33$ V, (**c**) $V_{udc}=34$ V and (**d**) $V_{udc}=35$ V. The upper harmonic voltage $V_{uac}$ is set to 0.1 V. The lower beam results are marked with blue lines and the upper beam results are marked with orange lines.

**Figure 13 micromachines-13-00989-f013:**
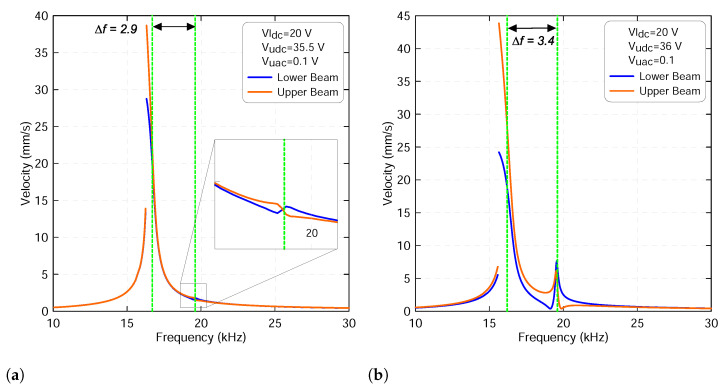
The frequency-response curves of the lower beam resonance frequency $f_{l1}$ and of the upper beam resonance frequency $f_{u1}$ with a lower voltage set to $V_{ldc}=20$ V and an upper voltage set to: (**a**) $V_{udc}=35.5$ V and (**b**) $V_{udc}=36$ V. The upper harmonic voltage $V_{uac}$ is set to 0.1 V. The lower beam results are marked with blue lines and the upper beam results are marked with orange lines.

**Table 1 micromachines-13-00989-t001:** Definition of the force expressions shown in the F.B.Ds.

	Lower Microbeam	Upper Microbeam
Normal Force	${\hat{F}}_{Nl}={\hat{N}}_{l}\;{\hat{w}}_{l}^{\prime \prime }$	${\hat{F}}_{Nu}={\hat{N}}_{u}\;{\hat{w}}_{u}^{\prime \prime }$
Damping Force	${\hat{F}}_{ld}=\hat{c}\;{\hat{\dot{w}}}_{l}$	${\hat{F}}_{ud}=\hat{c}\;{\hat{\dot{w}}}_{u}$
Mid-plane Stretching	$\frac{EA}{2\ell_{b}}\int_{0}^{\ell_{b}}\left({\hat{w}}_{l}^{\prime \prime }{{\hat{w}}_{l}^{\prime }}^{2}\right)dx$	$\frac{EA}{2\ell_{b}}\int_{0}^{\ell_{b}}\left({\hat{w}}_{u}^{\prime \prime }{{\hat{w}}_{u}^{\prime }}^{2}\right)dx$
Electrostatic Force	$\frac{\varepsilon b{\left(V_{ldc}+V_{lac}\cos\left(\hat{\Omega }\hat{t}\right)\right)}^{2}}{2{\left(d_{l}-{\hat{w}}_{l}\right)}^{2}}$	$\frac{\varepsilon b{\left(V_{udc}+V_{uac}\cos\left(\hat{\Omega }\hat{t}\right)\right)}^{2}}{2{\left(d_{u}+{\hat{w}}_{l}-{\hat{w}}_{u}\right)}^{2}}$

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
