# Peer review of "Investigation into Mode Localization of Electrostatically Coupled Shallow Microbeams for Potential Sensing Applications"

_micromachines, 2022, doi:10.3390/mi13070989_

Round 1
Reviewer 1 Report
The paper is devoted to a resonant sensor consisting of two electrostatically coupled microbeams. The authors describe their device as “a novel, smart and innovative”. Nonlinear and reduced-order models of the device are presented. In my opinion, this fully theoretical work is interesting for MEMS community and may be accepted for publication, but some practical information has to be included. Specifically, several questions have to be considered:
1. The described two-beam system is proposed as a sensor. However, there is no information on its sensitivity. I propose to estimate the sensitivity (for example, in terms of mass) and compare it with the literature data on the resonant sensors. This will confirm the advantages of the proposed design.
2. The authors write in the introduction, that “The design has the advantage of coupling electrostatically two microbeams serving as sensors to reduce the required input power and the device input power and footprint” (line 97). This statement is not obvious and requires explanation in the main text. It would be nice to see calculations and comparison with the existing devices.
3. The manuscript requires proofreading. There are a number of typos like “main advantageous”, line 32, “of te static”, line 317, “as clearly shown Figure 9”, line 362, etc.
Author Response
Dear Respected Reviewer,
Thank you very much for your helpful comments. These professional comments are indispensable for improving the quality of the paper. This paper has been carefully revised. And all the changes are highlighted in yellow in the revised manuscript. The following are our responses to the report comments.
Sincerely,
The authors
18/06/2022

Reviewer 2 Report
This paper introduces a new MEMS sensor design consisting of two microbeams coupled and excited electrostatically via one lower in-plane stationary electrode. And the dynamic characteristics of two coupled resonators are investigated around their respective primary resonances. The theoretical analysis of the paper is comprehensive. And the experiment is detailed and reasonable. But there are still some problems need to be illustrated. So the minor revision is necessary. The problems are given as follows:
1) The variables mentioned in the figure should be described in the text. Also, when describing the figure, necessary variables should to be marked in the figure. In addition, the font size of the text marked on the figure should be at least consistent with the text.
2) In Figure 5, the zero bias of ” the beam mid-point deflections of the lower beam and upper beam” changes a lot when the upper microbeam voltage is increased to 40 V. Wouldn't this have an adverse impact on the actual detection process?
3) In line 242, it says no crossover phenomenon occurs. But in Figure 6, it has the “crossover” mark on the figure. It's a bit contradictory.
4) In line 261, it says “the dynamic oscillations of the lower beam are much higher than that of the upper beam”. But in Figure 6, there are two resonance peak. And in the right one, the dynamic oscillations of the upper beam are higher than that of the lower beam.
5) At the beginning of the article, the author mentioned that this method can improve the sensitivity of detection. But from the Experimental results, the increase in sensitivity is not elaborated in quantify.
6) The full name of the same abbreviation should be consistent. And it only needs to be explained once in the text.
In summary, the work of this paper is worthy of publication. It provides a new solution for improving the performance of the sensor. And the use of two different beam’s configurations in one device showed possibility toward detecting two potential substances at the same time with two interacting distinct resonant peaks.
Author Response

(The authors gave the same response as above.)
